# NEUROMODULATION GATED TRANSFORMER

**Kobe Knowles, Joshua Bensemann, Diana Benavides Prado, Vithya Yogarajan,**
**Michael Witbrock, Gillian Dobbie, & Yang Chen**
School of Computer Science
University of Auckland
Auckland, New Zealand
`kobe.knowles@auckland.ac.nz`

## ABSTRACT

We introduce a novel architecture, the Neuromodulation Gated Transformer (NGT), which implements neuromodulation in transformers via a multiplicative effect. We compare it to baselines and show that it results in the best average performance on the SuperGLUE benchmark validation sets.

## 1 INTRODUCTION

Cellular neuromodulation is a biological mechanism involving neurons, where their intrinsic properties are continuously modified in a context-dependent manner according to stimuli, i.e., biochemicals called neuromodulators (Bargmann & Marder, 2013; Marder et al., 2014; Shine et al., 2021; Vecoven et al., 2020); it allows for the regulation of a population of neurons (Katz & Edwards, 1999). It has achieved notable success in the continual learning domain (Beaulieu et al., 2020; Ellefsen et al., 2015; Velez & Clune, 2017). Transformers (Vaswani et al., 2017) are architectures that eliminate recurrence by relying entirely on attention. The extensive developments in transformers have resulted in the monopolisation of the natural language processing and question answering (QA) leaderboards (Chowdhery et al., 2022; Fedus et al., 2022; Khashabi et al., 2022; Zoph et al., 2022).

The entwinement of neuromodulation and transformers is largely unexplored. We analyse the impact of integrating neuromodulation with the transformer via a multiplicative effect in non-continual learning scenarios. Specifically in QA on the SuperGLUE benchmark (Wang et al., 2019), which provides a range of language understanding tasks and a single metric to evaluate them all. We hypothesise that integrating neuromodulation with transformers will allow more complex data patterns to be learned, resulting in improved performance. The general idea is that the activations of a layer represent latent variables, which can act as context for other activations in the same layer. A block of parameters processes the output activations of a layer, producing an identical matrix of values between zero and one, which suppresses or enhances activations of the layer via the Hadamard product, relative to a value of 0.5.

Our preliminary experiments using NGT provide promising results. Adding neuromodulation improves performance on some datasets compared to baselines. Although individual datasets' performance varies, overall, it results in the best average performance on the validation sets.

## 2 NGT ARCHITECTURE

We introduce the Neuromodulation Gated Transformer (NGT), which is inspired by Beaulieu et al. (2020) and extends the work of Knowles (2022) by integrating a gating block with the transformer whose stimuli is entirely internal, eschewing external stimuli. See Figure 1 for an overview.

The gating block's purpose is to modify the intrinsic properties of output activations of a layer. The context is the other output activations of the layer, representing latent variables. The output of the gating block has the same dimensions as the input, producing values between zero and one, which dampen and enhance, respectively, the input activations (output activations of layer $k-1$).

Formally, the application of a gating block to layer $k-1$ is:

$$x_{k-1} = Layer_{k-1}(x_{k-2}), \; x_{gate} = GB(x_{k-1}) \odot x_{k-1}, \; x_k = Layer_k(x_{gate}), \qquad (1)$$

Figure 1: The NGT Transformer with one gating block.

where $x_{k-2}$ is the input to the $(k-1)$-th layer, $\odot$ is the Hadamard product, $GB$ is the gating block which consists of stacked transformer layers with a sigmoid function applied at the end, and $Layer_k$ is the $k$-th layer—we use BERT (Devlin et al., 2019) for our experiments. We note that multiple gating blocks can be applied simultaneously to different layers.

## 3 EXPERIMENTS

We evaluate the performance of NGT by comparing it to two baselines on the SuperGLUE benchmark. The NGT model is denoted by *neuromodulated-gating*, which contains a gating block with three layers. The gating block layers process the output activations of a layer and, via the attention mechanism in transformers, relate them with each other, producing values to enhance or suppress the activations conditioned on the context of other activations. The two baselines are: *non-neuromodulated-gating*, which is identical to neuromodulated-gating, except the output of the gating block is directed to the next layer; and *no-gating-block*, which is the unchanged model. Each model is based on the large variant of BERT, and after layer 21 is where all gating blocks are inserted. An independent model is trained for each dataset. Appendix A includes additional details on the experiments and models; Appendix B includes results comparing different positions of the gating block; Appendix C contains details regarding reproducibility.

Table 1: SuperGLUE benchmark validation sets results. A **bold** entry represents the best score for a dataset, and an underline represents the best score between the neuromodulated-gating and non-neuromodulated-gating models.

| Model | Metrics | no-gating-block | neuromodulated-gating | non-neuromodulated-gating |
|---|---|---|---|---|
| BoolQ | Acc. | 76.60±2.6 | **78.36±0.14** | 72.11±8.64 |
| CB | F1/Acc. | **87.86±1.30/87.50±1.79** | 82.44±5.41/85.12±4.49 | 85.81±4.07/85.12±1.03 |
| COPA | Acc. | 73.67±1.15 | **74.67±2.31** | 74.00±5.00 |
| MultiRC | $F1_a$/$EM_q$ | 64.25±5.68/13.26±12.38 | 70.22±0.41/23.22±1.07 | **70.68±0.29/24.45±1.21** |
| ReCoRD | F1/EM | **55.96±33.19/55.24±33.17** | 54.93±33.51/54.24±33.34 | 36.85±32.92/36.12±32.88 |
| RTE | Acc. | 74.13±0.21 | 72.32±0.84 | **74.37±2.73** |
| WiC | Acc. | **74.03±0.39** | 73.62±0.55 | 73.77±0.77 |
| WSC | Acc. | **65.70±2.22** | 65.06±0.55 | 64.74±1.11 |
| | Mean | 68.27±12.24 | **68.64±11.98** | 66.06±12.24 |

The results are displayed in Table 1. Neuromodulated-gating achieves the best performance averaged over all datasets; however, performance varies between datasets and no model is statistically significant (all p-values are $> 0.05$). We observe that no-gating-block achieves competitive performance, acquiring the best performance on half of the datasets. Surprisingly, non-neuromodulated-gating results in a worse average performance, although it contains three more layers than no-gating-block. This suggests that our approach to inserting randomly initialised parameters into a pre-trained model is not optimal. Better performance may be achieved by intermediate pre-training or pre-training a new model from scratch.

## 4 CONCLUSION

Overall, we find that adding neuromodulation results in an improved performance on the Super-GLUE benchmark validation sets compared to a model with the same number of parameters; however, performance across datasets varies and no result is statistically significant. Additionally, poor performance by non-neuromodulated-gating suggests that the potential of neuromodulation has not been reached. Further pre-training with the gating block will likely improve performance.

## URM Statement

Author Kobe Knowles meets the URM criteria of ICLR 2023 Tiny Papers Track.

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

## A EXPERIMENT DETAILS

Section A.1 describes the transformer architecture used in our experiments and Section A.2 outlines details regarding fine-tuning and the evaluation of a model's performance.

### A.1 TRANSFORMER ARCHITECTURE

We use the large BERT variant (Devlin et al., 2019) from the HuggingFace library (Wolf et al., 2020) as our pre-trained model. More specifically, we use the bert-large-cased variant. We extend BERT with the incorporation of a single gating block, which during our experiments consists of three layers identical in structure to the other layers of BERT—resulting in an additional 37,788,672 parameters. The gating layer's parameters are initialised to values from a normal distribution of mean zero and standard deviation 0.02; all layer normalization epsilon values are initialised to e-12; and all dropout layers have a dropout rate of 0.1. We briefly test whether inserting the gating block at the end position (layer 21) results in better performance than inserting it at the start position (layer 3). In all cases, inserting it at the end position results in better performance. For further details, see Appendix B.

Table 2: The number of parameters for models used in this paper. *There will be small deviations depending on how many output units are used for a dataset.

| Model | Parameters |
|---|---|
| no-gating-block | 333,580,289 |
| neuromodulated-gating | 371,368,961 |
| non-neuromodulated-gating | 371,368,961 |

In our experiments we compare three variants of architectures: an architecture with no gating block (*no-gating-block*); with a gating block and no gating, i.e., the gating block acts as additional layers (*non-neuromodulated-gating*); and with a gating block and gating (*neuromodulated-gating*). Table 2 showcases the total number of parameters for all models. To generate a prediction, at the classification token position, a single layer perceptron (SLP) is utilised. The output of the SLP will either be: a single value, where the sigmoid function is used to generate a probability; or multiple values, where the softmax function is used to generate probabilities.

### A.2 FINE-TUNING AND EVALUATION

In all experiments, every model is fine-tuned separately three times for each of the datasets in the SuperGLUE benchmark: BoolQ (Clark et al., 2019), CB (de Marneffe et al., 2019), COPA (Roemmele et al., 2011), MultiRC (Khashabi et al., 2018), ReCoRD (Zhang et al., 2018), RTE (Bar-Haim et al., 2006; Bentivogli et al., 2009; Dagan et al., 2006; Giampiccolo et al., 2007), WiC (Pilehvar & Camacho-Collados, 2019), and WSC (Levesque et al., 2012). Each of the three experiments has different factors that can change the results: the parameters of the gating block are randomly initialised and the training sets are shuffled uniformly at random in our experiments. When computing the mean for a dataset, if there are multiple metrics, e.g., F1-score and accuracy, they are averaged. For each dataset, the epoch corresponding with the highest metric score is chosen for each of the three experiments. The resulting metric scores are averaged and reported; this is done using the validation sets. We omit test set results because they require an online submission that has a monthly submission limit.

A single Quadro RTX 8000 GPU is used to conduct each experiment with a batch size of eight always being used. The Adam optimizer with weight decay (Kingma & Ba, 2014) is employed, where the hyperparameters $\beta_1$, $\beta_2$, and the weight decay rate are set to 0.9, 0.999, and 0.01, respectively; the initial learning rate is e-5 and is decayed via cosine decay (Loshchilov & Hutter, 2016) to a value of zero over the duration of training (i.e., 10 epochs); and each example is padded or truncated (from the start) to a length of 512. Cross-entropy loss is used for all datasets and WordPiece embeddings (Wu et al., 2016), through the pre-trained bert-large-cased tokenizer from HuggingFace (Wolf et al., 2020), are used to convert the tokens to vectors.

All reported results are converted to percentages and are rounded to two decimal places. The mean column in all tables is rounded to two decimal places and is calculated using the results already rounded to two decimal places themselves. The standard deviation—the square root of the variance of the three runs—is reported for all experiments; for the mean column this is the average of all reported standard deviations. We assume that each sample standard deviation is calculated using the same number of samples—all consist of three samples. Where there is multiple metrics we average their standard deviations and still assume the same number of samples; this is consistent with how the mean is calculated. We use $\sqrt{(x_1^2 + x_2^2 + \ldots + x_n^2)/n}$ to calculate the average standard deviation for the mean, where $x_n$ is the standard deviation for each dataset (averaged for multiple metrics) and $n$ is the number of datasets.

Table 3: Description of experiments. "LR" refers to learning rate, "BCE" to Binary Crossentropy, and CCE to Categorical Crossentropy. *We note that in our code this value is set to 9,999 and 4,999 (for 10,000 and 5,000, respectively) because the indexing starts at zero not one. ‡Each answer option is processed separately by the model; the answer option that generates the highest probability is the answer to the question.

| Experiment | Dataset | Epochs | Shuffle | Adam Optimizer | | | Start LR | Decay | | | Loss Function | Output Units |
|---|---|---|---|---|---|---|---|---|---|---|---|---|
| | | | | $\beta_1$ | $\beta_2$ | Decay Rate | | Function | LR | Steps | | |
| | BoolQ | 10 | T | 0.9 | 0.999 | 0.01 | e-5 | Cosine | 0 | 11,790 | BCE | 1 |
| | CB | 10 | T | 0.9 | 0.999 | 0.01 | e-5 | Cosine | 0 | 320 | CCE | 3 |
| | COPA | 10 | T | 0.9 | 0.999 | 0.01 | e-5 | Cosine | 0 | 1,000 | BCE | $1^{\ddagger}$ |
| Table 1 | MultiRC | 5 | T | 0.9 | 0.999 | 0.01 | e-5 | Cosine | 0 | 17,030 | BCE | 1 |
| | ReCoRD | 1 | T | 0.9 | 0.999 | 0.01 | e-5 | Cosine | 0 | 147,425 | BCE | 1 |
| | RTE | 10 | T | 0.9 | 0.999 | 0.01 | e-5 | Cosine | 0 | 3,120 | BCE | 1 |
| | WiC | 10 | T | 0.9 | 0.999 | 0.01 | e-5 | Cosine | 0 | 6,790 | BCE | 1 |
| | WSC | 10 | T | 0.9 | 0.999 | 0.01 | e-5 | Cosine | 0 | 700 | BCE | 1 |
| | BoolQ | 3 | T | 0.9 | 0.999 | 0.01 | e-5 | Cosine | 0 | 11,790 | BCE | 1 |
| | CB | 3 | T | 0.9 | 0.999 | 0.01 | e-5 | Cosine | 0 | 320 | CCE | 3 |
| | COPA | 3 | T | 0.9 | 0.999 | 0.01 | e-5 | Cosine | 0 | 1,000 | BCE | $1^{\ddagger}$ |
| Table 4 | MultiRC | 3 | T | 0.9 | 0.999 | 0.01 | e-5 | Cosine | 0 | 17,030 | BCE | 1 |
| | ReCoRD | 1 | T | 0.9 | 0.999 | 0.01 | e-5 | Cosine | 0 | 147,425 | BCE | 1 |
| | RTE | 3 | T | 0.9 | 0.999 | 0.01 | e-5 | Cosine | 0 | 3,120 | BCE | 1 |
| | WiC | 3 | T | 0.9 | 0.999 | 0.01 | e-5 | Cosine | 0 | 6,790 | BCE | 1 |
| | WSC | 3 | T | 0.9 | 0.999 | 0.01 | e-5 | Cosine | 0 | 700 | BCE | 1 |

Table 3 provides a description of the experiments run for each dataset of the SuperGLUE benchmark. Because ReCoRD and MultiRC are large datasets we only train them for 1 and 5 epochs, respectively. In all experiments Acc. refers to the accuracy, EM to exact-match accuracy, and F1 to the F1-score (i.e., the harmonic mean of precision and recall). For CB, the unweighted average of the F1 score per class is computed. For ReCoRD, the token-level F1 score is computed, with the entity that gives the max score being selected as the answer. For MultiRC, the F1 score is computed over all answer-options ($F1_a$) and the exact-match between each question's set of answers is computed ($EM_q$), i.e., all answers must be correct for a question for it to be counted as correct.

## B  IMPACT OF THE GATING BLOCK POSITION

Table 4 compares inserting the gating block in the earlier and later layers. The goal is to determine whether inserting the gating block at the start position (layer 3) or end position (layer 21) results in better performance after three epochs. We find that inserting the gating block at the end position results in better performance on all datasets.

The experiments are set up nearly identically to those in Section 3 and Appendix A with minor differences. Each model is trained for three epochs on each dataset except for ReCoRD, which because of the large dataset size is only trained for one epoch—for the end position the results are the same as reported in Table 1. The learning rate decay is left unchanged from those in Table 1, simulating the first three of ten epochs.

Table 4: Comparing the gating block at the start and end positions. All results are from the validation sets of the SuperGLUE benchmark. A **bold** entry represents the best score for a dataset. Each model is run for three epochs on each dataset with the exception of ReCoRD, which was trained for only one epoch.

| Model | Metrics | gating-start | gating-end |
|---|---|---|---|
| BoolQ | Acc. | 65.73±6.16 | **75.77±0.51** |
| CB | F1/Acc. | 47.35±2.16/67.86±3.09 | **77.96±5.38/82.14±3.57** |
| COPA | Acc. | 57.33±2.31 | **72.67±1.53** |
| MultiRC | $F1_a$/$EM_q$ | 45.44±39.37/14.41±12.38 | **70.19±0.18/22.91±2.12** |
| ReCoRD | F1/EM | 35.26±33.81/34.56±33.8 | **54.93±33.51/54.24±33.34** |
| RTE | Acc. | 62.21±5.51 | **72.20±1.25** |
| WiC | Acc. | 62.59±10.95 | **73.35±1.24** |
| WSC | Acc. | 62.5±2.54 | **63.46±0.00** |

## C    REPRODUCIBILITY

All code used to conduct the experiments is located in the following GitHub repository: `https://github.com/KobeKnowles/Neuromodulation-Gated-Transformer`.

