# OpenReview forum: "Neuromodulation Gated Transformer"
_ICLR.cc/2023/TinyPapers — Submitted to Tiny Papers @ ICLR 2023_

### Official Review · Reviewer_9WY1 · 2023-03-24

**Confidence:** 4

**Summary Of Contributions:**

The paper introduces neuromodulation in transformer through gating block.

**Rating:**

Clear, Correct, and Reproducible (CCR): a submission which meets the reviewing criteria

**Strengths And Weaknesses:**

The paper is well-written and provides a detailed description of the architecture and experimental setup. However, the paper could benefit from a more rigorous evaluation and analysis of the proposed NGT approach.

**Suggested Changes:**

While the paper includes some analysis and discussion of the results, it could benefit from going in-depth on the implications of the findings. The authors can consider including further ablation studies and testing on additional benchmarks. This would help to demonstrate the significance of the research and provide insights for future research in the field.

---

### Official Review · Reviewer_z4qP · 2023-04-02

**Confidence:** 3

**Summary Of Contributions:**

This work proposes a novel architecture which integrates neuromodulation in a transformer framework via a multiplicative effect in non-continual learning settings and analyzes its impact.

**Rating:**

Clear, Correct, and Reproducible (CCR): a submission which meets the reviewing criteria

**Strengths And Weaknesses:**

Clarity :
-this is a well-structured paper that has been written concisely
-findings are communicated clearly and effectively
-relevant literature has been included though this section can be improved.

Reproducibility: Code has been provided. Publicly available datasets have been used and instructions for downloading the data have been provided in the code repository.

Follows basic requirements: Yes

Correctness: The paper provides the technical details of the implementation. It is a good start and can be improved further by addressing these comments:

- for the ReCoRD dataset, the difference in the F1 and EM is relatively high between the neuromodulated and non-neuromodulated models (compared to the other datasets). Would the authors hypothesize any reasons for this ? (say running it for >1 epoch may reduce this disparity ? )
- conclusions: for further pre-training with gating block to improve performance - could a supervised fine-tuning approach using a % of the original dataset also help in improving performance ?
- only assessed on the validation set. Yes, the authors do mention about the monthly limit for the test-set evaluations, but test results for even 1-2 datasets would have helped immensely in understanding the performance of the proposed model.

**Suggested Changes:**

- Please see the comments above in the 'Correctness' sub-section.
- Regarding neuromodulation in QA, I came across the following paper: https://researchspace.auckland.ac.nz/bitstream/handle/2292/60957/Kobe-2022-thesis.pdf?sequence=1&isAllowed=y
Please add it to the references.

---

### Author Response · Authors · 2023-05-07
**Response to Reviewers**

We would like to thank all reviewers for their comments.

We have uploaded a revised version of the paper taking into account comments by the reviewers.

Unsurprisingly, the results are not statistically significant and have modified the paper to mention this, including a more careful conclusion.

We believe that there is a misunderstanding by the meta-reviewer in regard to their second point. We compare two baselines, one with fewer parameters (the unchanged bert model) and the other with the same number of parameters but no gating mechanism. Performance by NGT is better (not statistically significant) than the baseline with the same number of parameters. The inclusion of the unchanged bert model shows that our approach of inserting randomly initialised layers is sub-optimal, as shown by the worse performance by the baseline with more parameters (the same as NGT). It follows that the new parameters will likely have a detrimental effect in NGT without some form of intermediate fine-tuning/pre-training to calibrate the new parameters (which should be done in future work). We modify the conclusion to re-emphasise that the model is better than the baseline with the same number of parameters (which we believe should be the main focus as everything is identical except for the exclusion of the gating mechanism). Also, all models are trained (fine-tuned) identically on the same amount of data.

To make clear, we modify the paper to state that activations are suppressed/enhanced relative to a value of 0.5. We hypothesise that the additional capabilities provided by the gating block---the values of latent variables in a layer can influence and modify the values of other latent variables in that layer---will improve performance.

We cite the previous work mentioned and state that this work is an extension where the stimuli is entirely internal---there is no external stimuli like in the cited work.

We modify our description of neuromodulation to be more careful and fixate on describing cellular neuromodulation to narrow the focus.

On the ReCoRD dataset, the difference is due to the non-neuromoulated model converging on the majority class. This is not included due to the two page limit.

Lastly, we plan on uploading the revised paper to arXiv.

Thank you again for your comments.

---

### Author Response · Authors · 2023-05-29
**Opt-in for archival**

We wish to opt-in for archival.

---

### Meta-Review · Area_Chair_XtJj · 2023-04-04

**Recommendation:** Invite to archive
**Confidence:** 5

**Metareview:**

The reviewers praise the clarity of the paper and that the code is provided for reproducibility. I agree with this.

However, although both reviewers have given a CCR rating to this paper, I disagree: it is not "correct" in the sense defined by the guideline. In other words, the claims and conclusions are _not_ justified by the results.

The first reason for this disagreement is rather straightforward: the claim that the new model performs better is based on values that may not be statistically different from baseline. In Table 1, the errors are not defined (and should be) and the values have not been compared with statistical tests. Based on the magnitude of the error bars (± values), I would be surprised if the performance of the new model(s) turns out statistically different from that of BERT in any of the tasks. This should be done and reported in the paper.

I don't think new benchmarks are necessary in addition to what's already been used.

The second reason is a bit more involved, but has not been considered in the paper: to what extent do the improvements in the new model may arise from the new model(s) being larger and having used more compute (additional training)? The non-neuromodulated model's "higher" performance on some tasks (in quotes, due to lack of stats) in fact argues for this (that the exact architecture of the new layer isn't important---rather, that it exists and that the model was trained more after the addition of this layer is). These considerations should certainly be addressed.

The gating block seems to be scaling the output of layer 21. I am unclear how or why this should improve performance. Can't the next layer just adapt to this linearly scaled set of outputs? (the authors also say that the outputs are "suppressed" or "enhanced" but if the Hadamard product is with a matrix that's only between 0 and 1, wouldn't that only suppress the output, not enhance?).

The authors should revise the paper to include a brief discussion of the precise differences with the NeMoT model cited by reviewer `z4qP`.

Finally, neuromodulation is not introduced satisfactorily in that although the introduction touches on some aspects of it, it doesn't correctly or accurately describe neuromodulation. For example, neuromodulation is not unique to humans, nor is it restricted to behavior or learning. The term "neuromodulation" has also been used in BCI contexts recently, which is not how it's meant in the paper and this distinction is not made clear. The text should be edited to provide a more careful introduction to the concept.

**Summary:**

A new layer is added after layer 21 of a pretrained variant of BERT, which gates the output of each unit in layer 21 based on the activation of all other units in layer 21. The code is provided and paper is written rather clearly. Claims are not entirely supported by the evidence.

**Reason For Not Giving A Higher Recommendation:**

There main claims and conclusions are _not_ supported by the presented evidence. Another very similar piece of work has not at all been discussed.

**Reason For Not Giving A Lower Recommendation:**

Aspects of the core innovation are novel and are worth pursuing further. With revisions the paper can become CCR.

---

### Decision · Program_Chairs · 2023-04-07

Invite to archive